# Evaluation of the Abdala Vaccine: Antibody and Cellular Response to the RBD Domain of SARS-CoV-2

**DOI:** 10.3390/vaccines11121787

**Published:** 2023-11-30

**Authors:** Lorenzo Islas-Vazquez, Yan Carlos Alvarado-Alvarado, Marisa Cruz-Aguilar, Henry Velazquez-Soto, Eduardo Villalobos-Gonzalez, Gloria Ornelas-Hall, Sonia Mayra Perez-Tapia, Maria C. Jimenez-Martinez

**Affiliations:** 1Department of Immunology and Research Unit, Institute of Ophthalmology “Conde de Valenciana Foundation”, Mexico City 06800, Mexico; lorenzo.islas@institutodeoftalmologia.org (L.I.-V.);; 2Unidad de Vigilancia Epidemiológica Hospitalaria, Institute of Ophthalmology “Conde de Valenciana Foundation”, Mexico City 06800, Mexico; 3Unidad de Desarrollo e Investigación en Bioterapéuticos (UDIBI), Escuela Nacional de Ciencias Biológicas, Instituto Politécnico Nacional, Mexico City 11340, Mexico; 4Laboratorio Nacional para Servicios Especializados de Investigación, Desarrollo e Innovación (I+D+i) para Farmoquímicos y Biotecnológicos, LANSEIDI-FarBiotec-CONACyT, Mexico City 11340, Mexico; 5Departamento de Inmunología, Escuela Nacional de Ciencias Biológicas, Instituto Politécnico Nacional (ENCB-IPN), Mexico City 11340, Mexico; 6Department of Biochemistry, Faculty of Medicine, National Autonomous University of Mexico, Mexico City 04510, Mexico

**Keywords:** SARS-CoV-2, vaccine, COVID-19, Abdala vaccine, B lymphocytes, T lymphocytes, cytokines

## Abstract

Abdala is a recently released RBD protein subunit vaccine against SARS-CoV-2. A few countries, including Mexico, have adopted Abdala as a booster dose in their COVID-19 vaccination schemes. Despite that, most of the Mexican population has received full-scheme vaccination with platforms other than Abdala; little is known regarding Abdala’s immunological features, such as its antibody production and T- and B-cell-specific response induction. This work aimed to study antibody production and the adaptive cellular response in the Mexican population that received the Abdala vaccine as a booster. We recruited 25 volunteers and evaluated their RBD-specific antibody production, T- and B-cell-activating profiles, and cytokine production. Our results showed that the Abdala vaccine increases the concentration of RBD IgG-specific antibodies. Regarding the cellular response, after challenging peripheral blood cultures with RBD, the plasmablast (CD19+CD27+CD38^High^) and transitional B-cell (CD19+CD21+CD38^High^) percentages increased significantly, while T cells showed an increased activated phenotype (CD3+CD4+CD25+CD69+ and CD3+CD4+CD25+HLA-DR+). Also, IL-2 and IFN-γ increased significantly in the supernatant of the RBD-stimulated cells. Our results suggest that Abdala vaccination, used as a booster, evokes antibody production and the activation of previously generated memory against the SARS-CoV-2 RBD domain.

## 1. Introduction

The coronavirus disease 2019 (COVID-19) pandemic caused thousands of human deaths worldwide but also produced great successes in biomedical research with the rapid development of multiple vaccines against COVID-19 and their global application. COVID-19 vaccines have been proven safe and effective in reducing severe illness and death [1,2,3].

Although various studies carried out in different populations worldwide have demonstrated the short-term efficacy of vaccines against COVID-19, the duration of this protection over more extended periods requires continued study [4,5,6]. Since the approval of various vaccine products for global use by the World Health Organization, multiple studies have shown that the number of neutralizing antibodies induced by all the primary vaccine regimens decreases around 6–8 months after the last immunization, as does their clinical effectiveness [6,7,8].

Therefore, investigations on the effectiveness and duration of vaccination-induced immunity are essential to make relevant decisions about pandemic policy and to determine adequate timing for booster doses [9,10]. The booster consists of a recombinant protein subunit vaccine known as the Abdala vaccine (CIGB 66). This vaccine, developed by the Cuban Genetic Engineering and Biotechnology Centre, includes SARS-CoV-2′s recombinant protein receptor-binding domain (RBD), with aluminum hydroxide gel as an adjuvant. Abdala’s clinical trials, assessing its safety, immunogenicity, and efficacy, showed that it was well tolerated, and no severe adverse events were reported; therefore, its safe use as a primary scheme or as a booster was suggested [11,12,13]. In conjunction with the low cost of each vaccine dose and the low storage, refrigeration, and distribution requirements, these results led to the decision to use the Abdala vaccine as a booster against COVID-19 in the Mexican population [10].

Nevertheless, it is of interest to evaluate whether applying a protein-based vaccine booster would be equally effective in the Mexican population where different vaccine platforms have been administered. Therefore, this study aims to analyze the production of IgG (humoral immunity) after the application of a booster dose based on a recombinant protein subunit vaccine and the adaptative immune response (B and T lymphocytes and cytokines) against RBD protein in vitro, regardless of the initial vaccination scheme received.

## 2. Materials and Methods

### 2.1. Subjects Studied

We included samples from 25 volunteers (3 male and 22 female subjects), the average age was 39 years (range of 23–58 years), scheduled for vaccination with the Abdala vaccine. All participants had a complete scheme of vaccination and one booster dose. The last booster dose was applied, on average, 10 months before this study (in a range of 5–13 months). None of the participants mentioned having COVID-19 in the last six months, and post-vaccination symptoms were denied. The demographic data are shown in Table 1. Written informed consent was obtained from all participants enrolled in this study before peripheral blood collection. The Scientific (CI-055-2021), Biosecurity (CB-056-2021), and Bioethics (CEI-2020/10/09) Committees of the Institute of Ophthalmology “Conde de Valenciana Foundation” approved this study and were updated in February 2023.

### 2.2. Blood and Serum Sample Collection

In total, 10–12 mL of peripheral blood was obtained through venipuncture. Peripheral blood samples were collected before the Abdala vaccine (CIGB-66, recombinant protein including receptor binding domain (RBD) platform) dose, 14 ± 5 days after vaccine application, and after six months of follow-up (Figure 1A). To obtain the serum samples, the blood sample was allowed to clot for 10 min at room temperature and was centrifugated at 3500 rpm (1000× *g*) for 7 min. Then, the serum was aliquoted and stored at −20 °C until antibody consumption, and an enzyme-linked immunosorbent assay (ELISA) was performed. The blood sample obtained 14 ± 5 days post-Abdala vaccination was used immediately for a stimulation assay and flow cytometry analysis (Figure 1B).

### 2.3. Anti-RBD Antibody Consumption

In total, 200 μL of serum from pre-Abdala vaccination, 14 ± 5 days post-Abdala vaccination, and six-month-post-Abdala vaccination samples were incubated with RPMI medium with RBD recombinant protein (0.5 μg/mL) (UDIBI, IPN, Mexico City, Mexico) and with RPMI medium (Gibco, Paisley, Scotland, UK) without RBD, as a negative control (Neg Ctrl), for 24 h at 37 °C with 5% CO_2_, in a 24-well flat-bottom plate. After incubation, the serum was harvested and stored until the ELISA was performed.

### 2.4. Detection of IgGs Related to SARS-CoV-2 by Enzyme-Linked Immunosorbent Assay (ELISA)

IgGs related to SARS-CoV-2 against the RBD domain of S protein were detected in serum collected pre-, 14 ± 5 days post-, and six months post-Abdala vaccination, as well in serum harvested from the antibody consumption assay, using ELISA Kit UDITEST-V2G (IPN, Mexico City, Mexico) in accordance with the manufacturer’s instructions. The optical density (OD) was measured at 450 nm using the Multiskan Ascent spectrophotometer (Thermo Scientific, Waltham, MA, USA). Each sample was run in duplicate, and the values were averaged.

### 2.5. Stimulation Assay Using Whole Blood

In total, 400 μL of whole blood from 14 ± 5 days post-Abdala vaccination samples were stimulated in vitro with supplemented RPMI medium (10% fetal calf serum and 1% penicillin/streptomycin) with RBD recombinant protein (0.5 μg/mL) (UDIBI, IPN, Mexico City, Mexico), with supplemented RPMI medium (10% fetal calf serum and 1% penicillin/streptomycin) without RBD as a negative control (Neg Ctrl), or with supplemented RPMI medium (10% fetal calf serum and 1% penicillin/streptomycin) with PMA–ionomycin (30 ng/mL and 0.3 mg/mL, respectively) as a positive control (Pos Ctrl), for 24 h at 37 °C with 5% CO_2_ in a 24-well flat-bottom plate. After 24 h, the supernatants in each condition were harvested and stored until further use for cytokine detection. Also, the cells were recovered for flow cytometric analysis.

### 2.6. Cytometric Bead Array (CBA)

The cytokines IL-2, IL-4, IL-6, IL-10, IL-17, TNF-α, and IFN-γ were detected in supernatants from each stimulation assay condition with the Th1, Th2, and Th17 cytometric bead array (CBA) (BD Bioscience, Franklin Lakes, NJ, USA) in accordance with the manufacturer’s instructions. Data were acquired using the BD FACSLyrics cytometer (BD, San José, CA, USA) and BD FACSuite software Version 1.4. Then, the data were analyzed, employing FCAP Array software version 3.0 (BD San José, CA, USA).

### 2.7. Multi-Parametric Flow Cytometry Analysis

The following labeled anti-human monoclonal antibodies (MoAbs) were used for the B-lymphocyte analysis: FITC anti-CD19 (HIB19 clone, BD, #555412), PE anti-CD21 (B-ly4 clone, BD, #555422), APC-H7 anti-CD27 (M-T271 clone, BD, #560222), and PE-Cy7 anti-CD38 (HB-7, Biolegend, #356608). On the other hand, for helper-T-lymphocyte analysis, V605 anti-CD3 (HIT3a clone, BD, #564712), APC anti-CD4 (RPA-T4 clone, BD, #555349), PE-Cy5 anti-CD25 (M-A251 clone, BD, #555433), PE anti-CD69 (FN50 clone, Biolegend, #310906), and PE-Cy7 anti-HLA-DR (LN3 clone, Tonbo, #60-9956-T100) were used.

Briefly, 100 μL of recovered cells was placed in a cytometric tube, and anti-CD19, anti-CD21, anti-CD27, and anti-CD38 MoAbs or anti-CD3, anti-CD4, anti-CD25, anti-CD69, and anti-HLA-DR MoAbs were added. The tubes were incubated at room temperature for 30 min. After incubation, 1 mL of BD FACS Lysing solution (BD, San José, CA, USA) was added, incubated at room temperature for 12 min, washed, and centrifugated. Finally, the cellular pellet was resuspended and fixed with 300 μL of BD Stabilizing Fixative solution (San José, CA, USA).

Events were acquired using the BD FACSLyrics (San José, CA, USA) cytometer and using the BD FACSuite software Version 1.4. First, cells were gated in an FSC-A vs. FSC-H dot plot to exclude doublets. Then, the lymphocyte region was selected using an FSC vs. SSC dot plot; 20,000 events were acquired from this population. Next, an SSC-A vs. CD19 dot plot or an SSC-A vs. CD3 followed by CD3 vs. CD4 dot plot was made. From the gated-CD19 region, a CD27 vs. CD21 plot, a CD21 vs. CD38 plot, and a CD27 vs. CD38 plot were used to identify and quantify the resting-memory (CD21+CD27+) and effector-memory (CD21-CD27+) B-lymphocyte populations, transitional (CD21+CD38^high^) B lymphocytes, and plasmablast cells (CD19+CD27+CD38^high^), respectively. Conversely, from the gated-CD3+CD4+ region, a CD4 vs. CD25 plot, CD25 vs. CD69 plot, and CD25 vs. HLA-DR plot were used to identify and quantify the activated helper T lymphocytes (Figure 2). Positive and negative gates for each molecule were verified based on the “fluorescence-minus-one (FMO)” control (Appendix A).

### 2.8. Statistical Analysis

For parametric distribution, the Shapiro–Wilk test was used. Statistical analysis was performed using the Wilcoxon test for paired samples and the Mann–Whitney test to compare two unpaired groups. All results are shown as the mean ± standard deviation (SD). Analyses were performed using the statistical program Graph Pad Prism 8 (GraphPad Software, La Jolla, CA, USA). *p*-values of <0.05 were considered statistically significant.

## 3. Results

### 3.1. Abdala Vaccine Increases the OD of Specific IgGs against RBD after Two Weeks of Vaccination and Maintains It over Six Months

Since the main objective of vaccination is the production of antibodies against SARS-CoV-2, we analyzed the OD due to the production of antibodies by the Abdala vaccine, and we detected a significant increase two weeks post-Abdala vaccination compared to that in the pre-Abdala vaccination samples (**1.178 vs. 1.046, respectively; *p* < 0.0001**); meanwhile, after six months of follow-up, the OD was similar to that post-Abdala vaccination (Figure 3A). Also, no differences were detected when analyzing the OD in terms of the initial vaccination scheme or in terms of the first booster dose (Appendix A). Previously, we reported changes in the OD with the first two doses of vaccination (Pfizer, Astra-Zeneca, and Sputnik V) in the Mexican population [14]. In addition, we analyzed samples from subjects with the booster dose and from subjects with at least ten months on average since their booster dose. When comparing the OD two weeks post-Abdala vaccination, we detected that it was similar to that observed in the samples obtained after the second dose (**1.155 vs. 1.178, respectively**) but was slightly higher than the OD of the booster dose samples and of the subjects at ten months since their booster dose (Figure 3B). Therefore, we can suggest that the level of antibodies was maintained throughout the vaccination scheme with Pfizer, Astra Zeneca, and Sputnik V, and that the Abdala vaccine could provide humoral protection against SARS-CoV-2.

### 3.2. Consumption of IgGs before and after Abdala Vaccination and Six Months Post-Abdala Vaccination

The efficacy of the vaccines is based on the production of anti-RBD antibodies capable of specifically recognizing the RBD SARS-CoV-2 protein domain. In this sense, we incubated the RBD recombinant protein with serum collected before and 14 ± 5 days after Abdala vaccination and six months post-Abdala vaccination to detect differences in OD, due to antibody binding to recombinant RBD protein, resulting in antibody consumption. When analyzing the OD after incubation with the RDB recombinant protein, a significative decrease in OD was detected in both the pre- and 14 ± 5 days post-Abdala vaccination samples, as well as the six-month-post-Abdala vaccination samples compared to that of the negative control **(*p* < 0.0001, *p =* 0.0025, and *p* < 0.0001, respectively)**. However, it should be mentioned that the decrease was lower in the 14 ± 5 days post-Abdala vaccination samples compared to that in the pre-Abdala vaccination samples **(*p* < 0.0001)** (Figure 4). These results may suggest that the antibodies could recognize the RBD protein with distinct affinities.

### 3.3. RBD Stimulation Induces Plasmablast and Transitional B-Lymphocyte Increases

Besides the generation of antibodies, another central feature of vaccination is to generate immunological memory that can protect in the long term and respond when the antigen is present again. Previously, we analyzed and reported the proportion of memory and activated-memory B lymphocytes throughout the vaccination schedule, and no differences were detected between the first and second dose [14]. Therefore, we induce a response in the post-Abdala vaccination blood samples by exposing them to the RBD recombinant protein to simulate the re-encounter with the viral protein. After the stimulus, we analyzed the proportion of resting- and activated-memory B lymphocytes (CD19+CD21+CD27+ and CD19+CD21−CD27+, respectively), as well as the proportion of plasmablast cells (CD19+CD27+CD38^high^) and transitional B lymphocytes (CD19+CD21+CD38^high^).

After the stimulus, no changes were detected in the proportion of resting-memory B lymphocytes, while a significant decrease was observed in the proportion of activated-memory B lymphocytes (**3.315% vs. 2.694%, *p =* 0.0063**) (Figure 5A). Conversely, in the proportion of plasmablast cells, a significative increase was detected after RBD stimulation (**2.372% vs. 2.734%, *p* = 0.0034**) (Figure 5B). Finally, the proportion of transitional B lymphocytes increased when the RDB protein was added (**1.486% vs. 2.569%, *p* < 0.0001**) (Figure 5C). These results suggest that the B lymphocytes generated along the immunization scheme retain the capability to activate in response to the RBD protein, supporting long-term protection.

### 3.4. RBD Stimulation Induces Helper-T-Lymphocyte Activation, as well as IL-2 and IFN-γ Production

Like B lymphocytes, helper T lymphocytes are essential cells in long-term protection. Therefore, we also analyzed the proportion of helper T lymphocytes that responded to the RBD protein, employing the expression of CD25, CD69, and HLA-DR. A slight non-significant increase in the proportion of CD25+ helper T lymphocytes (CD3+CD4+CD25+) was detected (**12.7% vs. 13.27%, *p =* 0.2493**) (Figure 6A). However, when comparing the proportions of CD25+CD69+ and CD25+HLA-DR+ helper T lymphocytes after stimulation, a significative increase was detected (**0.4480% vs. 0.8457%** (***p =* 0.0008**) and **5.497% vs. 6.372%** (***p =* 0.0012**), respectively) (Figure 6B,C). These results suggest that helper T lymphocytes can be activated in response to the RBD protein, supporting an anti-viral response.

It is known that activated helper T lymphocytes’ effector function is the production of cytokines to help other immune cells, so to confirm the response to the RBD protein, we analyzed the production of IL-2 and IFN-γ. We collected the supernatant from the stimulus and quantified the IL-2 and IFN-γ concentrations. A significative increase in both IL-2 and IFN-γ was detected (**3.015 pg/mL vs. 12.21 pg/mL** (***p* < 0.0001**) and **3.955 pg/mL vs. 13.76 pg/mL** (***p* < 0.0001**), respectively) (Figure 7A,B). Overall, these results support the fact that helper T lymphocytes induced throughout the distinct immunizations have the capability to respond to the RBD protein and to perform their effector function.

## 4. Discussion

Vaccines against SARS-CoV-2 have been proven to be highly effective in neutralizing the virus, thus reducing both the number of cases and the number of deaths worldwide, because the vaccines induce a robust, effective, and protective immune response based mainly on the production of neutralizing antibodies against SARS-CoV-2.

The early and effective development of COVID-19 vaccines based on several platforms resulted in a high demand for their global purchase, so some countries decided to develop their own vaccines to make this treatment more accessible to their population and to reduce the cost of acquiring other vaccines, as is the case of Cuba’s development of the Abdala vaccine. However, there are few reports on the effectiveness of this vaccine in generating immune protection or enhancing pre-existing immune memory against SARS-CoV-2 when used as a booster dose [11,12,13].

Recently, the Abdala vaccine was included as part of the booster scheme in the Mexican population, and we decided to analyze whether this vaccine could induce antibodies against SARS-CoV-2. We followed the healthcare staff of the Institute of Ophthalmology “Conde de Valenciana Foundation” throughout the vaccination schedule, and the OD of IgG related to SARS-CoV-2 remained at similar levels from the second vaccine dose. Two weeks after Abdala vaccine’s application, the OD increased as a consequence of inoculation; this means that the vaccine is able to induce antibodies against SARS-CoV-2, as reported in clinical trials with non-vaccinated subjects [11]. However, as the staff were previously vaccinated, the increase detected in our study may be because the Abdala vaccine stimulates the pre-existing immune memory generated by previous vaccine doses [14].

Concentration and antibody affinity are fundamental characteristics for the efficacy of vaccines, and the latter determines the antibodies’ ability to detect the RDB protein domain. Measuring the anti-RBD antibodies in the serum is commonly used to determine prior exposure to SARS-CoV-2 infection and to assess the anti-viral protection capacity. Some reports have analyzed the affinity of the antibodies generated in convalescent patients and by distinct vaccinal platforms and detected that the antibodies present a distinct affinity for the RBD protein and variants such as Omicron [15,16,17]. In this sense, in our antibody consumption assay, the difference in the decreased OD detected could be due to the affinity of the antibodies induced by the Abdala vaccine to binding the recombinant RBD protein. The results are similar to those of Muruato et al., wherein a fluorescence-based assay demonstrated neutralizing activity against SARS-CoV2 through a reduction in fluorescence comparable to that of a plaque reduction neutralizing assay [18].

It has been reported that an early humoral immune response is mediated mainly by plasmablasts, which produce antibodies with a weaker affinity to the specific antigen compared to plasma cells (as we discussed later). After 6 months, the time course for the affinity maturation of anti-RBD antibodies could explain the changes in the OD at this time compared to the post-Abdala vaccination time [19,20]. These data could provide information about the humoral immune response of vaccinated individuals, and the follow-up could determine seroprevalence.

In addition to the production of neutralizing antibodies, the induction and activation of the lymphocyte-mediated adaptive immune response is required to effectively control and eliminate viral infections. After vaccination, the immune system retains a memory ability that protects against subsequent infections and prevents disease progression to a severe stage. We analyzed the percentage of memory B lymphocytes after a stimulus with the RBD recombinant protein, and a decrease in activated-memory B lymphocytes and an increase in the percentage of transitional B lymphocytes (CD19+CD21+CD38^High^) and plasmablast cells (CD19+CD27+CD38^High^) were detected. This phenomenon might be evidence that the presence of RBD can stimulate activated-memory B lymphocytes, which, in an early response against an antigen, can differentiate into antibody-producing cells identified as plasmablasts in the short term. It has been reported that plasmablasts are cells with short lives that produce antibodies with less affinity than plasmatic cells. In several studies, B lymphocytes activated by SARS-CoV-2 mRNA vaccines and the adenoviral vector [19,20,21,22,23,24,25,26] differentiate into plasmablasts and increase their percentage within 7 to 10 days after the start of infection or post-vaccination, producing large amounts of neutralizing antibodies to eliminate the pathogen, which, however, decrease after viral elimination or one month after vaccination [19,21,22,23,24,25,26]. This could indicate that plasmablasts leave before the germinal center develops and have a lower antigen affinity [26]. Since the population of plasmablasts is transitory, maintaining high levels of plasmablasts is possible only by recruiting new virgin B lymphocytes, self-renewing plasmablasts, or, in the case of the vaccine, restimulating memory B lymphocytes and, particularly, activating memory B lymphocytes [23].

Meanwhile, in our study, the presence of transitional B lymphocytes could explain why, in this timeframe (24 h), a small proportion of activated-memory B lymphocytes started the differentiation process into plasmatic cells. Taken together, the reason the respective antibodies were detected could be because we analyzed samples only two weeks after the Abdala dose, when short-lived plasmablast cells could be contributing to plasma antibodies and even activated-memory B lymphocytes could be differentiated and still expanding to plasmatic cells [19,20]. Conversely, six months after vaccination, the plasma cells would be the main source of antibodies, resulting in affinity maturation that decreases the OD in the antibody consumption assay.

On the other hand, several studies focused on the adaptive immune response following a SARS-CoV-2 infection and post-vaccination and reported that the increase in CD4+ T lymphocytes is associated with the control of primary SARS-CoV-2 infection via a decrease in both the symptoms and severity of the disease and via rapid viral elimination. Also, an increase in the percentage of CD4+ T lymphocytes after each dose was reported [27,28,29]. Therefore, it is vital to mention that a T cell response is essential in viral infections, and in this sense, we analyzed whether the presence of the RBD protein can induce activation and cytokine production related to T lymphocytes, as well as IL-2 and IFN-γ.

Activation was defined as the expression of CD25 and the co-expression of CD25 with CD69 and of CD25 with HLA-DR. An increase in CD25+CD69+ T lymphocytes would be evidence of early activation; meanwhile, an increase in CD25+HLA-DR+ T lymphocytes would indicate an effector function. It has been previously shown that the expression of HLA-DR, CD38, and Ki-67 reliably characterizes human T cells recently activated in vivo during acute and chronic viral infections [28]. Also, the production of cytokines such as IL-2 and IFN-γ is evidence of the effector function of T-lymphocyte subpopulations.

The production of cytokines in COVID-19 patients and CD4+ T lymphocytes isolated from SARS-CoV-2-vaccinated subjects stimulated with a peptide pool predominantly expressed IL-2 and IFN-γ, whereas IL-17A, IL-4, and IL-10 were less frequent and not significant [30,31,32,33]. In our study, a significative increase in IL-2 and IFN-γ was detected; these cytokines are mainly secreted by helper 1 T lymphocytes (Th1), which could be responsible for cell-mediated immunity.

In addition, we detected a significative increase in IL-6, TNF-α, and IL-10, while a slight increase in IL-4 and IL-17 was observed (Appendix A). IL-4 and IL-10 are mainly secreted by helper 2 T lymphocytes (Th2), which can induce significant antibody production. On the other hand, higher levels of IL-6 and TNF-α would indicate the participation of innate immune cells such as monocytes/macrophages.

The present study has some limitations, and the results must be interpreted cautiously. For example, most subjects were females; it is well known that the humoral response is slightly higher in females than in males [34]. However, the higher representation of females in hospital staff reflects the distribution of persons in the context of our study. Another limitation was the determination of OD instead of IgG titers; it is essential to examining the neutralizing capacity of IgG to comprehend the efficacy of vaccines. Since the ELISA kit utilized in this study is semi-quantitative, antibody titers were not determined; nevertheless, neutralizing antibodies were detected via the assay [35]. Viral neutralization was not assessed in this investigation; it is crucial to comprehend that clinical immunity against various diseases or even mortality hinges on viral neutralization [13]. In addition, more assays are necessary to ascertain the affinity of the antibodies generated by the Abdala vaccination. However, at this time, this is not feasible due to the fact that the majority of the population has already received other COVID-19 vaccines, and subjects who have been vaccinated previously would have a combination of antibodies produced by the different doses received.

The results in this study suggest that the Abdala booster dose induces adaptive immune response activation, resulting in titers of antibodies similar to those of the initial vaccination scheme doses with other SARS-CoV2 vaccines [14]. However, it is important to note that a limiting factor related to the Abdala booster’s effectiveness is the presence of new variants of the SARS-CoV-2 virus that are not covered with this type of vaccine. Thus, individuals who have received the Abdala vaccination may face a potential risk of infection if they encounter a variant not targeted by this vaccine. Therefore, it is imperative to consistently observe the progression of SARS-CoV-2 mutations and, if needed, redesign approved vaccines or boosters that target these novel strains. Finally, our findings can be used as a first approach to demonstrate the effects of the Abdala vaccine on the immune system when used as a booster dose.

## 5. Conclusions

The Abdala vaccine induced specific anti-RBD IgGs two weeks post-vaccination and maintained them for six months after vaccination; the B and T lymphocytes generated throughout the vaccination scheme were present in the vaccinated subjects and maintained their capability to respond against the virus protein. Therefore, this vaccine could be employed as a booster dose in the Mexican population with the complete scheme and even with one or two booster doses. It is necessary to follow up on the antibody concentration and the antigen-specific response to establish a schedule for subsequent Abdala booster applications and effectiveness in the presence of new SARS-CoV-2 variants.

## Figures and Tables

**Figure 1 vaccines-11-01787-f001:**
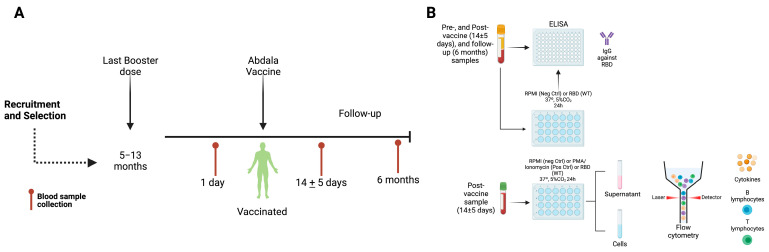
**Experimental design.** (**A**) Recruitment and selection of participants. Blood sample collection was conducted 1 day before Abdala vaccination, 14 ± 5 days post-Abdala vaccination, and after 6 months of follow-up, as indicated by the red marker. (**B**) Serum samples from pre-vaccination, 14 ± 5 days after post-Abdala vaccination and after 6 months of follow-up were employed for antibody consumption and ELISA. On the other hand, the blood sample collected in sodium heparin, 14 ± 5 days after Abdala vaccination, was employed for a stimulation assay and flow cytometry analysis. Created with BioRender.com.

**Figure 2 vaccines-11-01787-f002:**
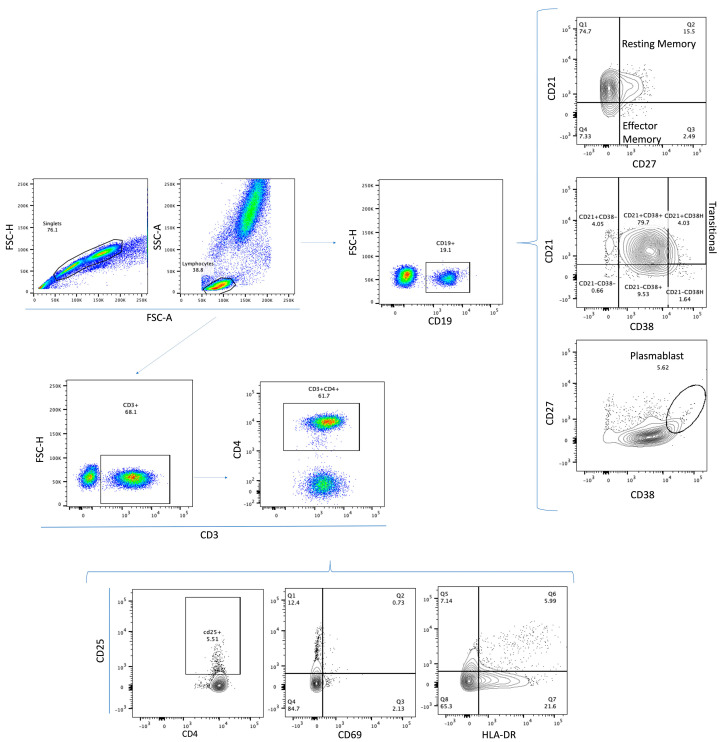
**Representative flow cytometry analysis.** In an FSC-A vs. FSC-H dot plot, we excluded doublets. Then, the lymphocyte region was selected in an FSC-A vs. SSC-A dot plot. Next, an SSC-A vs. CD19 dot plot was made. From the gated-CD19 region, a CD27 vs. CD21 plot, a CD38 vs. CD21 plot, and a CD38 vs. CD27 plot were made. Conversely, an SSC-A vs. CD3 followed by CD3 vs. CD4 dot plot was made. From the gated-CD3+CD4+ region, a CD4 vs. CD25 plot, CD25 vs. CD69 plot, and CD25 vs. HLA-DR plot were made.

**Figure 3 vaccines-11-01787-f003:**
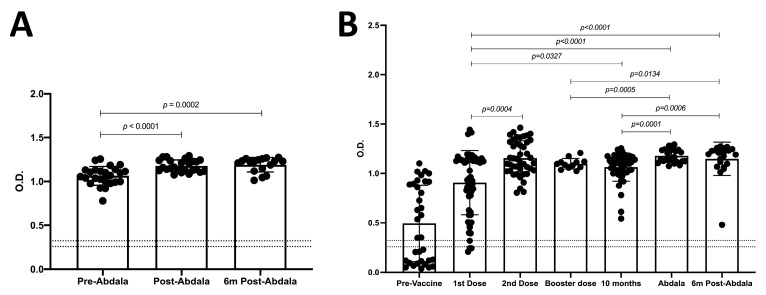
**IgGs related to SARS-CoV-2 against RBD domain of S protein induced at distinct points in the vaccination schedule.** (**A**) OD of IgG against RBD pre- and 14 ± 5 days post-Abdala vaccination and 6 months (6 m) post-Abdala vaccination. Pre-Abdala vaccination, *n* = 25; 14 ± 5 days post-Abdala vaccination, *n* = 25; and 6 months (6 m) post-Abdala vaccination, *n* = 17. (**B**) OD throughout vaccination scheme. Pre-vaccination (non-vaccinated), n = 34; first dose, *n* = 54; second dose, *n* = 44; booster dose, *n* = 12; 10 months after booster dose, *n* = 46; Abdala vaccination, *n* = 25; and 6 months (6 m) post-Abdala vaccination, *n* = 17. Cut-off values are indicated by lines. Mean and standard deviation (SD) are shown. Central trend values and dispersion values are indicated in Appendix A.

**Figure 4 vaccines-11-01787-f004:**
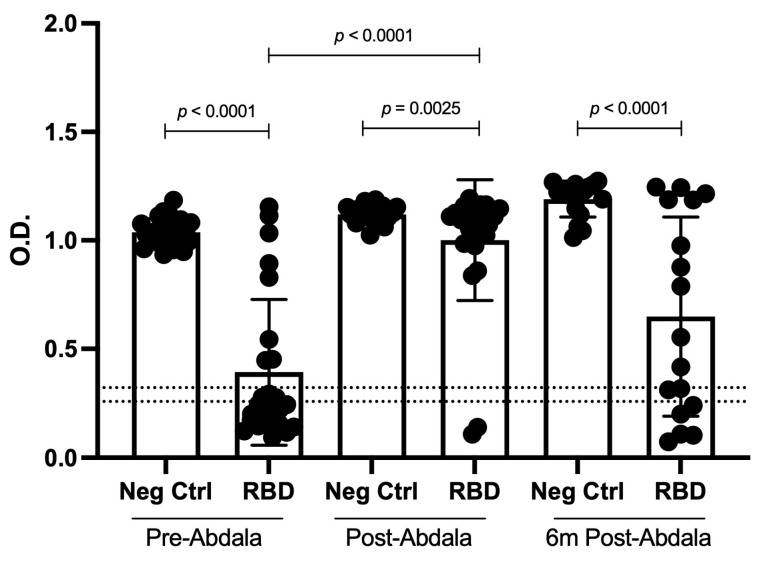
**Antibody consumption after 24 h incubated with and without RBD recombinant protein.** OD of IgG against RBD pre- and 14 ± 5 days post-Abdala vaccination and six months post-Abdala vaccination. Pre-Abdala vaccination, *n* = 25; 14 ± 5 days post-Abdala vaccination, *n* = 25; and 6 months (6 m) post-Abdala vaccination, *n* = 17. Cut-off values are indicated by lines. Mean and standard deviation (SD) are shown. Central trend values and dispersion values are indicated in Appendix A.

**Figure 5 vaccines-11-01787-f005:**
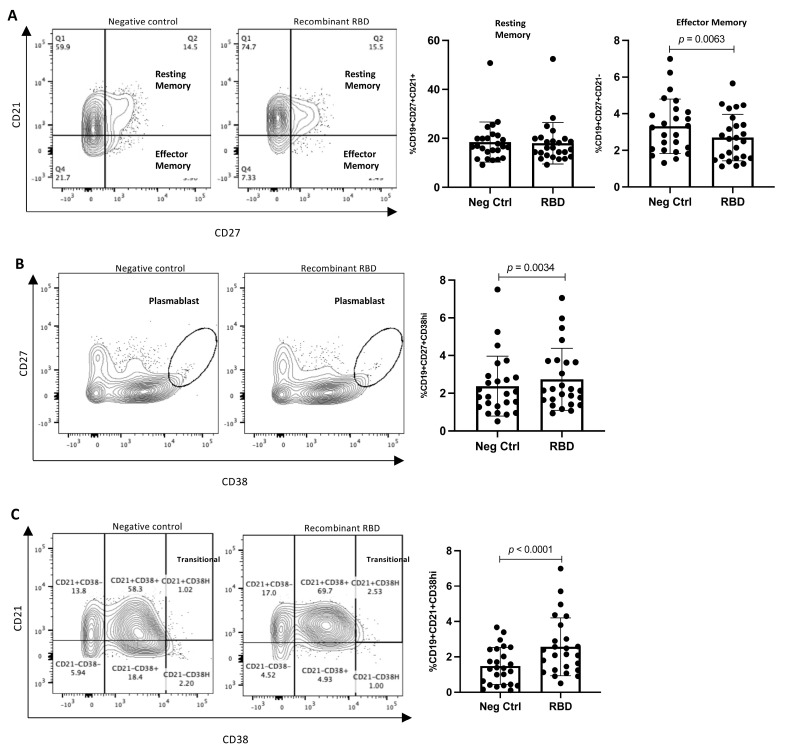
**Analysis of B lymphocytes after stimulus with and without RBD protein for 24 h.** (**A**) Percentage of resting-memory (CD19+CD21+CD27+) and activated–memory (CD19+CD21 − CD27+) B lymphocytes. (**B**) Percentage of plasmablast cells (CD19+CD27+CD38^high^). (**C**) Percentage of transitional B lymphocytes (CD19+CD21+CD38^high^). *n* = 25. Mean and standard deviation (SD) are shown. Central trend values and dispersion values are indicated in Appendix A.

**Figure 6 vaccines-11-01787-f006:**
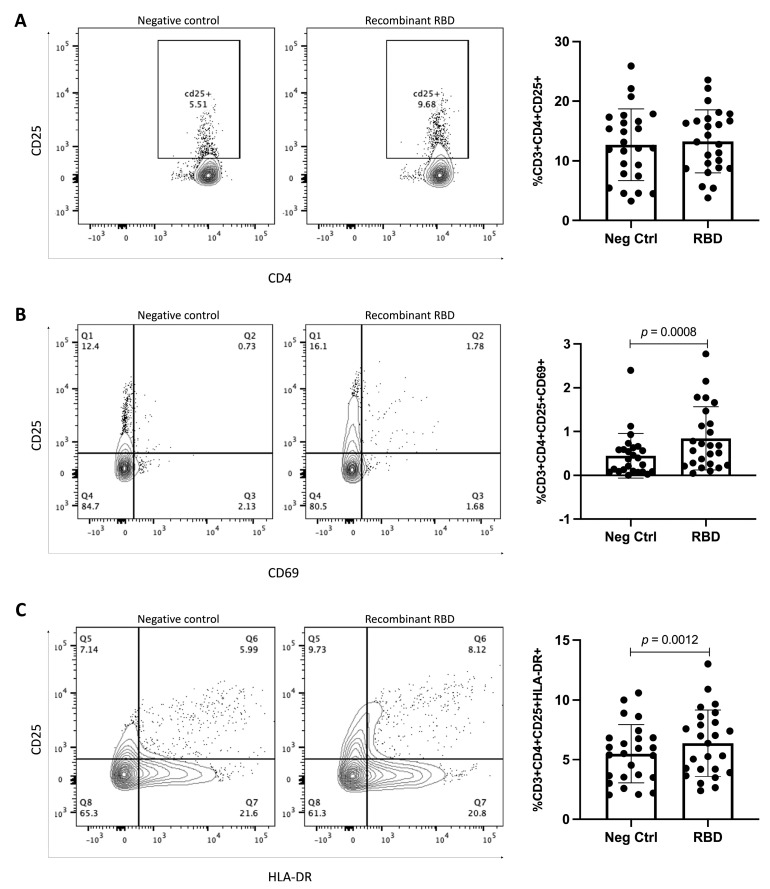
**Analysis of T lymphocytes after stimulus with and without RBD protein for 24 h.** (**A**) Percentage of CD3+CD4+CD25+ T lymphocytes gated on CD3+. (**B**) Percentage of CD3+CD4+CD25+CD69+ T lymphocytes gated on CD3+CD4+. (**C**) Percentage of CD3+CD4+CD25+ HLA-DR+ T lymphocytes gated on CD3+CD4+. *n* = 25. Mean and standard deviation (SD) are shown. Central trend values and dispersion values are indicated in Appendix A.

**Figure 7 vaccines-11-01787-f007:**
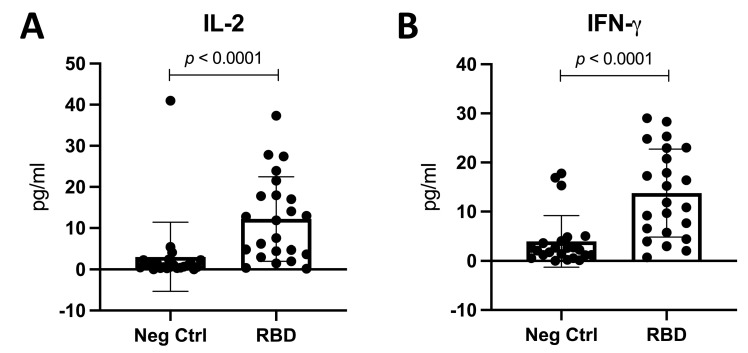
**Concentration of IL-2 and IFN-γ after stimulus with RBD protein for 24 h.** (**A**) pg/mL of IL-2. (**B**) pg/mL of IFN-γ. *n* = 23. Mean and standard deviation (SD) are shown. Central trend values and dispersion values are indicated in Appendix A.

**Table 1 vaccines-11-01787-t001:** Demographic data.

	Pre-Abdala Vaccine	Post-Abdala Vaccine	Six Months Post-Abdala Vaccine
n	25	25	17
Sex Male/Female	3/22	3/22	3/14
Age: Years (Range)	40 (25–58)	40 (25–58)	39 (25–58)
COVID-19 History (Last 6 month)	0	0	0
Last Booster Dose: Months (Range)	10 (5–13)	N/A	6 (6)
AEFI ^1^	N/A	0	N/A
Initial scheme:			
Pfizer-BioNTech (BNT162b2)	5	5	3
Astra Zeneca (AZD1222)	17	17	11
Sputnik V (Gam-COVID-Vac)	3	3	3
First Booster dose:			
Pfizer-BioNTech (BNT162b2)	3	3	2
Astra Zeneca (AZD1222)	12	12	7
Sputnik V (Gam-COVID-Vac)	10	10	8

^1^ AEFI: adverse events following immunization. N/A: not applicable.

## Data Availability

Data are contained within the article and Appendix A.

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
