# Peer review of "Evaluation of the Abdala Vaccine: Antibody and Cellular Response to the RBD Domain of SARS-CoV-2"

_vaccines, 2023, doi:10.3390/vaccines11121787_

Round 1

Reviewer 1 Report

Comments and Suggestions for Authors

There are several shortcomings and scientific mistakes in different sections:

  1. Abstract: The abstract lacks a clear statement of the research's significance and objectives. It briefly mentions that the Abdala vaccine induces an immune response, but it doesn't specify the novelty or broader implications of the findings. An abstract should provide a concise summary of the study's key points and significance to the field.

  2. Introduction: The introduction starts well by discussing the global context of COVID-19 and the need for booster doses. However, it lacks proper citations for the claims made, making it challenging for readers to verify the information. Furthermore, it fails to clearly define the specific objectives of the study, leaving the reader uncertain about the research's focus.

  3. Materials and Methods: The materials and methods section is informative. However, there are several issues:

    • The text formatting is inconsistent, making it challenging to follow the section's flow.
    • Some information about the study population, such as gender distribution and average age, is presented before the section header "2.1. Population studied," which is confusing and should be placed within the appropriate subsection.
    • The description of the antibody consumption assay is somewhat unclear. It needs more detailed information about the procedure and controls used.
    • The description of the flow cytometry analysis is quite detailed, but it can be challenging for readers unfamiliar with the techniques to follow. A more simplified and clear explanation might be beneficial.
  4. Discussion: While the discussion provides some insights into the findings, there are various issues:

    • The language is sometimes convoluted, making it difficult to understand the key points. For instance, the section discussing plasmablasts and affinity maturation is quite complex and could be simplified for a broader audience.
    • The discussion does not provide a proper context for the results, such as comparing the findings to other COVID-19 vaccines or explaining their implications for public health.
    • The last paragraph introduces the potential limitation of the Abdala vaccine in dealing with new SARS-CoV-2 variants but does not elaborate on this issue or discuss the broader implications.
  5. Conclusions: The conclusion could benefit from a more concise summary of the key findings and their relevance to the field of COVID-19 vaccination.

Comments on the Quality of English Language

The manuscript contains multiple typographical errors, inconsistent formatting, and missing or incomplete references to prior research, which need to be addressed.

Reviewer 2 Report

Comments and Suggestions for Authors

To investigate whether Abdala vaccine can be used to boost protection against SARS-CoV-2, the authors analyzed anti-RBD antibody responses and cellular responses after boost immunization of subjects who were previously vaccinated by other platforms than Abdala.

Anti-RBD IgG slightly but significantly increased after Abdala boosting. However, no data was presented about neutralizing activity or protection capacity for infection and disease progression. Therefore, the clinical significance of Abdala boosting is not clear and further researches are required.

In experiments of in vitro cellular responses, control experiments were not adequate so that I cannot agree with their conclusion that memory B- and T-lymphocytes present in vaccinated subjects maintain the capability to respond against virus protein.

Major problems:

(1) B and T cell responses in vitro

In sections 3.3 and 3.4, B and T cell responses to antigen in vitro were analyzed. When B and T cells were stimulated with recombinant RBD protein, they differentiated into activated states. These results indicated that there were RBD-specific memory B and T cells. However, it was not clear whether Abdala vaccine contributed to maintain memory cells. These memory cells could be established and maintained even without boosting. In order to reveal the vaccine effects in maintenance of immunological memory, it is necessary to compare B and T cells before and after (or with and without) Abdala boosting.

(2) Subjects

In the line 73 of the text, three males and 22 females were included in the study. However, in the Table1, they were three males and 21 females. Which is correct? In addition, why the gender ratio is unbalanced, and does it potentially influence the results? 

If the aim of the study was evaluation of the effect by booster dose of Abdala regardless of the regimen of initial vaccination, information about initial vaccination would be necessary, i.e. ratios of subjects who were received Pfizer, Astra-Zeneca, and Sputonik V, or one or two booster doses. And were there any differences between the groups of different initial vaccinations?

(3) Stimulation assay in whole blood

400 μL of whole blood were in vitro stimulated with RBD recombinant protein (line 116). If it is correct, the cell cultures contained serum proteins such as antibodies and cytokines, which might influence the downstream assays. Or whole blood cells without serum were cultured?

And it is not clear whether “post-Abdala vaccine samples” were collected at 14 days or 6 months after boosting.

(4) ELISA

It is better to show not OD but titers. (OD ~1 seems be the ceiling value.) And it is also important that produced IgGs have neutralizing capacities. If the authors showed such data, the paper would be more attractive.

(5) antibody consumption assay

The decrease of OD was lower in 14d post-Abdala samples (Fig. 4). The authors consider that it was because lower affinity of IgG produced by plasmablast which were developed without germinal center reactions, while affinity matured memory cells were activated but still in the process of differentiation into plasma cells. However, the period of two weeks would be enough for memory B cells to differentiate into plasma cells, and I think this interpretation is unreasonable.

Minor points:

Characters in figures were so small that it is difficult to read, especially in Figs. 1B, 2, 3B, 5.

Line 364, what does “title” mean?

Reviewer 3 Report

Comments and Suggestions for Authors

The article written by Islas-Vasquez et al., and entitled "Evaluation of Abdala Vaccine: Antibody and Cellular Response 2 to RBD Domain of SARS-CoV-2" is well presented and easy to understand. Mehtods are wellpresented and results are interesting and apapted to the conclusion of the work.

The main objective of this work is clear and it is to study the antibody production and adaptive cellular response in the Mexican population who received the Abdala vaccine as a booster. Results obtained are with interest. They demonstrated that the Abdala vaccine induced specific anti-RBD IgG's after two weeks post-vaccination and maintained over six months after vaccination, memory B- and T-lymphocytes present in vaccinated subjects maintain the capability to respond against virus protein.

However, some details should be clarified in the body text to improve the scientific sound of the article for researchers which are interested by the vaccine booster schema:

* Informations and details from the Abdala subunit vaccine platform should be added in the introduction and in Methods to give audiance the opportunity to understand this kind of vaccine.

* Fig 1/A should be improved by adding information of what happen at days 14/- 5 days and 6 months.

* In conclusion, authors should add the limitations of their study...

Comments on the Quality of English Language

The article written by Islas-Vasquez et al., and entitled "Evaluation of Abdala Vaccine: Antibody and Cellular Response 2 to RBD Domain of SARS-CoV-2" is well presented and easy to understand. Mehtods are wellpresented and results are interesting and apapted to the conclusion of the work.

The main objective of this work is clear and it is to study the antibody production and adaptive cellular response in the Mexican population who received the Abdala vaccine as a booster. Results obtained are with interest. They demonstrated that the Abdala vaccine induced specific anti-RBD IgG's after two weeks post-vaccination and maintained over six months after vaccination, memory B- and T-lymphocytes present in vaccinated subjects maintain the capability to respond against virus protein.

However, some details should be clarified in the body text to improve the scientific sound of the article for researchers which are interested by the vaccine booster schema:

* Informations and details from the Abdala subunit vaccine platform should be added in the introduction and in Methods to give audiance the opportunity to understand this kind of vaccine.

* Fig 1/A should be improved by adding information of what happen at days 14/- 5 days and 6 months.

* In conclusion, authors should add the limitations of their study...

Round 2

Reviewer 1 Report

Comments and Suggestions for Authors

The manuscript has been improved significantly.

Author Response

We appreciate your comments to improve this work.

Reviewer 2 Report

Comments and Suggestions for Authors

The authors carefully answered to my questions, and most of my concerns were cleared.

I am still not convinced about their explanation of antibody consumption assay. Memory B cells are generated through affinity maturation pathway in germinal centers which are induced by the initial vaccine scheme and first booster dose. Therefore, the affinity of plasmablasts differentiated from memory B cells is higher than that of naive B cells. I think it is not correct that plasmablast generated from memory B cells have low affinity to the antigen. Anyway, as the authors added in discussion, more experiments are required to clarify the affinity of antibodies produced by Abdala vaccination. To understand the results of antibody consumption assay, it is necessary to compare antibodies produced in primary vaccination and boostings.  

These two errors should be corrected.

Line 143. (HB-7, Biolegend, #XX), #XX should be replaced with the product number.

Line 383. Not Supplementary Figure 2, but Supplementary Figure 3.

Figs. 5 and 6 were invisible.
